# Consumers' psychological constructs regarding hybrid meat products: A scoping review protocol

**Jasmijn de Veld**[1], **Ceren Pekdemir**[2]*, **Gill ten Hoor**[1]*

**1** Department of Work and Social Psychology, Faculty of Psychology and Neuroscience Maastricht University, The Netherlands, **2** Maastricht Sustainability Institute, School of Business and Economics, Maastricht University, The Netherlands

These authors contributed equally to this work.
* ceren.pekdemir@maastrichtuniversity.nl (CP); gill.tenhoor@maastrichtuniversity.nl (GtH)

## Abstract

### Objective

The objective of this scoping review protocol is to outline the approach for mapping existing evidence on consumers' psychological constructs and outcomes related to hybrid meat products.

### Introduction

Rising global meat consumption exacerbates environmental and health challenges, while many consumers struggle to reduce their intake through plant-based alternatives alone. Hybrid meat products, combining plant-based and animal-derived components, offer a promising transition pathway toward more sustainable protein consumption. Although research on consumers' psychological constructs and outcomes to hybrid meat products is growing, a comprehensive overview is lacking.

### Inclusion criteria

The review will use the Population, Concept, Context (PCC) framework, focusing on consumers of any demographic who engage with hybrid meat products. It will include studies that examine at least one psychological construct (e.g., attitude) or outcome (e.g., acceptance) across various settings. Studies conducted in relevant contexts, such as retail environments, restaurants, or experimental settings, will be eligible for inclusion.

### Methods

The review will follow the Joanna Briggs Institute (JBI) methodology and will be reported in accordance with PRISMA-ScR guidelines. A structured search of all relevant publications up to December 2025 will be carried out across multiple databases via EBSCOhost and PubMed. Eligible peer-reviewed English and Dutch articles will

**Data availability statement:** All available data will be made publicly available on the Open Science Framework (OSF; DOI: 10.17605/OSF.IO/8UWVS; accessible at https://osf.io/mhpgf).

**Funding:** The author(s) received no specific funding for this work.

**Competing interests:** The authors have declared that no competing interests exist.

be selected based on predefined inclusion criteria. Titles, abstracts, and full texts will be screened sequentially, with reasons for exclusion recorded. Data will be extracted using a tailored extraction tool, with all modifications documented in the final review. The analysis will comprise frequency counts and a descriptive summary of key characteristics and themes, presented in tabular form.

## Introduction

Meat consumption is a major contributor to climate change, with food production systems responsible for 30% of global greenhouse gas emissions [1]. The anticipated growth in global meat demand [2] raises additional concerns regarding resource use and sustainability [3]. Beyond environmental implications, high meat consumption is associated with several adverse health outcomes, including cardiovascular diseases, type 2 diabetes, and certain types of cancer [4]. It also contributes to the global obesity epidemic, prompting urgent calls to reduce meat consumption worldwide [4].

Despite these concerns, many consumers remain unmotivated or unsuccessful in reducing their intake of animal-based meat by substituting it with plant-based alternatives [5]. One promising approach to bridge the gap between plant-based and animal-derived products is hybrid meat; a product that combines animal meat with plant-based or alternative protein sources in varying proportions [5,6]. Hybrid meat offers not only a potential pathway toward reduced meat consumption but also opportunities for innovation that could reshape the food system and facilitate the acceptance of cultivated meat [7].

Interest in hybrid meat products is increasing [8]. For example, an online survey of 2,405 consumers in Denmark, Spain, and the United Kingdom found that at least 57% of respondents were willing to try hybrid meat products, and 46% were willing to purchase them [9]. However, studies investigating the factors influencing this growing interest have yielded mixed results. Several studies identify taste and sensory attributes as primary determinants of consumer appeal [6,10–12]. In particular, hybrid meat products may achieve higher sensory acceptance compared with fully plant-based alternatives [8]. On the other hand, Grasso and Jaworska [6] found that negative evaluations were often driven by poor sensory quality rather than the hybrid concept itself. Beyond sensory characteristics, positive attitudes toward hybrid meat have also been linked to perceived sustainability and nutritional benefits [6,11]. Nevertheless, Banovic, Barone [10] reported that consumers' environmental self-identity and health consciousness had little influence on their attitudes, suggesting that such personal values do not necessarily translate into acceptance.

Several reviews have examined consumer attitudes, perceptions, and behaviours toward protein alternatives [13] and meat analogues more broadly [14]. One recent review addressed consumer attitudes toward hybrid meat products, focusing on studies published between 2020 and 2022 [8]. However, no comprehensive review has yet investigated psychological constructs or outcomes related to hybrid meat consumption. Psychological constructs, defined as unobservable concepts that integrate

patterns of thought, emotion, and behaviour into coherent units [15], are critical for understanding consumer motivations and designing effective communication and marketing strategies aimed at reducing meat consumption [14]. Given the expanding yet inconsistent literature, there is a clear need to systematically map existing evidence, clarify psychological drivers and barriers, and identify knowledge gaps. Moreover, comparing findings across socio-demographic groups and national contexts is essential for understanding broader market dynamics [6]. Insights from such work can inform targeted strategies to promote reduced meat consumption, for instance, by aligning interventions with populations or regions where consumption is highest or projected to increase [2]. Therefore, this scoping review aims to provide a comprehensive overview of existing evidence on consumers' psychological constructs and outcomes toward hybrid meat products. The findings will help guide future research, inform product development, and support evidence-based policy design.

## Materials and methods

This scoping review protocol is developed in accordance with the Preferred Reporting Items for Systematic review and Meta-Analysis Protocols (PRISMA-P) checklist [16] (See S1 Appendix). The proposed scoping review will be conducted following the methodological framework of the Joanna Briggs Institute (JBI) for scoping reviews [17] and reported in line with the Preferred Reporting Items for Scoping Reviews (PRISMA-ScR) guidelines [18].

### Review question

The scoping review question will be guided by the following research question, formulated according to the Population–Concept–Context (PCC) framework (see Table 1): *What is known about consumers' psychological constructs and outcomes related to hybrid meat products?*

### In- and exclusion criteria

The review will include peer-reviewed articles published in English or Dutch, regardless of publication year, to ensure a comprehensive overview of both foundational and contemporary studies. The inclusion and exclusion criteria will be informed by the PCC- framework, as outlined in Table 1.

Studies focusing solely on fully plant-based or lab-grown meat alternatives without a hybrid element will be excluded. Similarly, studies examining only producer or retailer perspectives will be considered only if they also address aspects of consumer behaviour. Technical or food science studies without consumer data will not be included. Only peer-reviewed studies presenting original empirical data will be eligible; consequently, grey literature, conference abstracts, editorials, and purely narrative papers will be excluded. The scoping review will consider analytical observational studies, including prospective and retrospective cohort studies, case-control studies, and analytical cross-sectional studies. Descriptive observational designs, such as case series, individual case reports, and descriptive cross-sectional studies, will also be eligible for inclusion. Qualitative research that explores qualitative data will be included, encompassing methodologies such as phenomenology, grounded theory, ethnography, qualitative description, action research, and feminist research.

**Table 1. Population–concept–context (PCC) framework.**

| Type | Description |
|---|---|
| Population | Consumers of any age, gender, or demographic background. This includes individuals who are actual or potential purchasers, users, or evaluators of hybrid meat products. |
| Concept | At least one psychological construct or outcome is measured. This includes, but is not limited to: behaviour; attitudes and perceptions; willingness to pay and purchase intention; sensory expectations and taste preferences; health, environmental, and ethical concerns; well-being; responses to marketing, labelling, packaging, and branding strategies, specifically related to hybrid meat products, defined as food products combining meat with plant-based or alternative protein sources. |
| Context | All relevant contexts: retail, restaurants, experimental settings, and different geographic/cultural contexts. |

Additionally, both experimental and quasi-experimental study designs, such as randomised controlled trials, non-randomised controlled trials, pre-post studies, and interrupted time series analyses, will be considered.

## Search strategy

The search strategy will follow the three-step approach recommended in the JBI methodology for scoping reviews [17]. First, a preliminary search was conducted in MEDLINE (PubMed) in June 2025, chosen for its extensive coverage of literature in nutrition, health, and behavioural sciences, as well as its advanced search functionality and integration with related databases [19]. This initial search focused on the concept of the product ("hybrid meat") to ensure that no relevant studies were overlooked. The concept of psychological constructs was not included in the search terms due to its broad and diverse nature, which cannot be effectively captured through keyword searches. Instead, articles will be assessed for psychological constructs during the screening phase. Titles, abstracts, and keywords of retrieved records were examined to identify relevant index terms and synonyms. To supplement this process, ChatGPT was consulted to identify additional terminology and variations associated with hybrid meat products, using the prompt *'Generate database-appropriate synonyms and related terms for "hybrid meat" that can be used in a search string for both PubMed and EBSCOhost.'*. The proposed synonyms were reviewed by the research team and incorporated into the search strategy where appropriate. New terms were cross-checked against MeSH. Based on this analysis, a comprehensive search strategy was developed (see S2 Appendix). The Boolean operator OR was used to combine synonymous terms and closely related concepts within the search strategy. MeSH terms were used where available to improve search precision. No filters or date restrictions will be applied to avoid the inadvertent exclusion of relevant studies. The final search will be conducted in January 2026 of all relevant publications up to December 2025 across multiple databases accessed via EBSCOhost, including APA PsycInfo, APA PsycArticles, Psychology and Behavioral Sciences Collection, CINAHL, Global Health, GreenFILE, and MEDLINE (PubMed). A cross-database adoption strategy will be used to ensure a comprehensive and systematic identification of relevant literature. The preliminary search strategy will be applied without modification during the final search. The reference lists of all included articles will be manually screened to identify additional relevant studies. These will be assessed at the full-text level to determine eligibility.

## Study selection

All retrieved records will be imported into Microsoft Excel for deduplication. Titles will be screened against the predefined inclusion criteria by two independent reviewers (see Table 1). Potentially eligible articles will be subsequently screened at the abstract level and, if relevant, at the full-text level by two reviewers. Reasons for exclusion at the full-text stage will be systematically recorded according to PRISMA-ScR guidelines. Common exclusion codes will include: absence of psychological constructs (PSY), lack of relevance to hybrid meat (HM), insufficient data (DT), duplicate record (DUP), language not meeting inclusion criteria (LNG), or inaccessible full text (TXT). Any disagreements between reviewers will be resolved through discussion until consensus is reached. A narrative summary of the selection process will be provided, accompanied by a PRISMA-ScR flow diagram to ensure transparent reporting. The record screening is expected to be completed by January 2026.

## Data extraction

Data from included studies will be extracted by two independent reviewers using a customised data extraction tool developed by the research team in Microsoft Excel (see S3 Appendix). Key data items will include: (1st) author, publication year, country of study, study design, study aim, participant characteristics, product type, psychological construct(s), main findings, and reported research gaps. The data extraction tool may be refined iteratively throughout the review process. Any modifications will be documented and justified in the final review report. Discrepancies will be resolved through discussion or, if necessary, consultation with an additional reviewer. When required, study authors will be contacted for clarification or to obtain missing information. Data extraction is expected to be completed by February 2026.

## Data analysis and presentation

Data will be analysed descriptively, in line with JBI and PRISMA-ScR recommendations. A frequency analysis will summarise study characteristics, including country, design, and methodology. A qualitative descriptive synthesis will then summarise key constructs, product types, findings, and identified research gaps. Emergent themes will be identified through thematic grouping of psychological constructs and outcomes. Results will be presented narratively and in tabular form to provide a clear and comprehensive overview of the existing evidence base. We anticipate that the study findings will be available in April 2026.

## Supporting information

**S1 Appendix. PRISMA-P 2015 checklist.**
(DOCX)

**S2 Appendix. Search Strategy.**
(DOCX)

**S3 Appendix. Data Extraction Form.**
(DOCX)

## Author contributions

**Conceptualization:** Jasmijn de Veld, Ceren Pekdemir, Gill ten Hoor.

**Data curation:** Jasmijn de Veld, Ceren Pekdemir, Gill ten Hoor.

**Formal analysis:** Jasmijn de Veld, Ceren Pekdemir, Gill ten Hoor.

**Methodology:** Jasmijn de Veld, Ceren Pekdemir, Gill ten Hoor.

**Supervision:** Ceren Pekdemir, Gill ten Hoor.

**Validation:** Jasmijn de Veld, Ceren Pekdemir, Gill ten Hoor.

**Writing – original draft:** Jasmijn de Veld, Ceren Pekdemir, Gill ten Hoor.

**Writing – review & editing:** Jasmijn de Veld, Ceren Pekdemir, Gill ten Hoor.

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
