## [Decision Letter · Decision Letter 0]

7 Jan 2026

Dear Dr. Pekdemir,

Thank you for submitting your manuscript to PLOS ONE. After careful consideration, we feel that it has merit but does not fully meet PLOS ONE’s publication criteria as it currently stands. Therefore, we invite you to submit a revised version of the manuscript that addresses the points raised during the review process.

We look forward to receiving your revised manuscript.

Kind regards,

António Raposo

Academic Editor

PLOS One

Journal Requirements:

3. Please include captions for your Supporting Information files at the end of your manuscript, and update any in-text citations to match accordingly. Please see our Supporting Information guidelines for more information: http://journals.plos.org/plosone/s/supporting-information .

4. We are unable to open your Supporting Information file “PRISMA-P checklist - seperate file.docx”. Please kindly revise as necessary and re-upload.

Reviewer's Responses to Questions

**Comments to the Author**

1. Does the manuscript provide a valid rationale for the proposed study, with clearly identified and justified research questions?

Reviewer #1: Yes

Reviewer #2: Yes

2. Is the protocol technically sound and planned in a manner that will lead to a meaningful outcome and allow testing the stated hypotheses?

Reviewer #1: Yes

Reviewer #2: Partly

3. Is the methodology feasible and described in sufficient detail to allow the work to be replicable?

Reviewer #1: Yes

Reviewer #2: No

4. Have the authors described where all data underlying the findings will be made available when the study is complete?

Reviewer #1: Yes

Reviewer #2: No

5. Is the manuscript presented in an intelligible fashion and written in standard English?

Reviewer #1: Yes

Reviewer #2: Yes

You may also provide optional suggestions and comments to authors that they might find helpful in planning their study.

Reviewer #1: Dear Authors,

The manuscript presents a well-defined scoping review protocol, addressing a current and relevant topic in light of the environmental and health challenges associated with meat consumption. The introduction is clear, well-structured, and contextualizes hybrid meat products as a promising strategy for transitioning to more sustainable dietary patterns, demonstrating theoretical consistency and good articulation with existing literature.

As suggestions for methodological improvement, in the methods section, note that the manuscript presents separate subtopics, such as research question, inclusion and exclusion criteria, and types of sources. We suggest that these contents be kept integrated into the larger topic "Methods," in accordance with the recommendations of the Joanna Briggs Institute. Furthermore, the content currently presented in the subtopic "Types of sources" could be directly incorporated into the inclusion criteria, contributing to greater clarity and organization of the text. It is also recommended that the manuscript include at least the main descriptors used in a standardized way in the search strategies, even if the complete strategies are available in supplementary material, in order to strengthen the transparency and reproducibility of the protocol. Finally, it is suggested to include page numbering corresponding to the items of the PRISMA-P checklist, facilitating the monitoring of compliance with reporting guidelines.

Overall, this is a methodologically consistent protocol, well aligned with international guidelines for scoping reviews and with the potential to offer a relevant contribution to the field. With minor adjustments to the organization and presentation of the method, the manuscript is likely to be further strengthened.

Reviewer #2: The use of AI is explicitly stated, which is positive in terms of transparency. However, the description is insufficiently detailed to guarantee reproducibility: It does not explain how ChatGPT was used (general prompt, iterative refinement, validation by experts). Furthermore, it is unclear whether the terms suggested by the AI were checked against controlled vocabularies (MeSH, APA Thesaurus, CINAHL Headings).

There is a potential incongruity: a preliminary search in June 2025 and a final search scheduled for January 2026. It would be important to clarify whether the strategy will be updated or revalidated before the final search, especially considering the use of emerging terms in an innovative field such as hybrid foods.

Although it is mentioned that the complete strategy is in Appendix 2, the main text could indicate: the use of Boolean operators; the searched fields (ti, ab, kw); and the cross-database adaptation strategy.

**Do you want your identity to be public for this peer review?** For information about this choice, including consent withdrawal, please see our Privacy Policy

Reviewer #1: **Yes:** Rodrigo Assis Neves Dantas Dantas

Reviewer #2: No

---

## [Author Response · Author response to Decision Letter 1]

26 Jan 2026

Dear Reviewers,

Thank you very much for your feedback. We hereby respond to the points raised:

We have rechecked the PLOS ONE style requirements and confirm that we fully meet the PLOS ONE’s style requirements now.

All available data will be made publicly available on the Open Science Framework. We have received a confirmation on January 26, 2026 from Plos ONE that the information was updated on our behalf. We have received the instruction to not include it in the Manuscript itself as the data availability statement will be published alongside our manuscript should it be accepted for publication.

We have added the heading “Supporting Information” to the manuscript. Furthermore, we have updated the captions of the supporting materials to S1 Appendix: PRISMA-P 2015 Checklist, S2 Appendix: Search Strategy, and S3 Appendix: Data Extraction Form. We also updated the in-text citations accordingly to refer to S1 Appendix, S2 Appendix, and S3 Appendix.

4. We are unable to open your Supporting Information file “PRISMA-P checklist - seperate file.docx”. Please kindly revise as necessary and re-upload.

The updated PRISMA P-checklist is now attached in PDF format in the appendix.

Not applicable.

The reference list has been reviewed and no additional changes were made.

Reviewer #1:

Dear Authors,

- The manuscript presents a well-defined scoping review protocol, addressing a current and relevant topic in light of the environmental and health challenges associated with meat consumption. The introduction is clear, well-structured, and contextualizes hybrid meat products as a promising strategy for transitioning to more sustainable dietary patterns, demonstrating theoretical consistency and good articulation with existing literature.

Thank you very much.

- As suggestions for methodological improvement, in the methods section, note that the manuscript presents separate subtopics, such as research question, inclusion and exclusion criteria, and types of sources. We suggest that these contents be kept integrated into the larger topic "Methods," in accordance with the recommendations of the Joanna Briggs Institute.

Thank you very much for your suggestions. We changed this as suggested.

- Furthermore, the content currently presented in the subtopic "Types of sources" could be directly incorporated into the inclusion criteria, contributing to greater clarity and organization of the text.

We have deleted the sub-heading ‘types of sources’, so that this section is directly incorporated into the inclusion criteria.

- It is also recommended that the manuscript include at least the main descriptors used in a standardized way in the search strategies, even if the complete strategies are available in supplementary material, in order to strengthen the transparency and reproducibility of the protocol.

In order to strengthen the transparency and reproducibility of the protocol the following sentences have been added to the text:

‘’The Boolean operator OR was used to combine synonymous terms and closely related concepts within the search strategy.’’

‘’MeSH terms were used where available to improve search precision.’’

‘’A cross-database adoption strategy will be used to ensure a comprehensive and systematic identification of relevant literature.’’

- Finally, it is suggested to include page numbering corresponding to the items of the PRISMA-P checklist, facilitating the monitoring of compliance with reporting guidelines.

This is done in the updated PRISMA P-checklist attached in the appendix. See also our response to the editor.

- Overall, this is a methodologically consistent protocol, well aligned with international guidelines for scoping reviews and with the potential to offer a relevant contribution to the field. With minor adjustments to the organization and presentation of the method, the manuscript is likely to be further strengthened.

Thank you very much.

Reviewer #2:

- The use of AI is explicitly stated, which is positive in terms of transparency. However, the description is insufficiently detailed to guarantee reproducibility: It does not explain how ChatGPT was used (general prompt, iterative refinement, validation by experts). Furthermore, it is unclear whether the terms suggested by the AI were checked against controlled vocabularies (MeSH, APA Thesaurus, CINAHL Headings).

Thank you very much for your feedback. We have detailed the description of using ChatGPT by including the following text:

‘’To supplement this process, ChatGPT was consulted to identify additional terminology and variations associated with hybrid meat products, using the prompt ‘Generate database-appropriate synonyms and related terms for “hybrid meat” that can be used in a search string for both PubMed and EBSCOhost.’. The proposed synonyms were reviewed by the research team and incorporated into the search strategy where appropriate. New terms were cross-checked against MeSH.’’

- There is a potential incongruity: a preliminary search in June 2025 and a final search scheduled for January 2026. It would be important to clarify whether the strategy will be updated or revalidated before the final search, especially considering the use of emerging terms in an innovative field such as hybrid foods.

Thank you very much for bringing this point to our attention. The strategy remained unchanged for the final search, as we believe it remains current and appropriate for the field. For clarity, we have added the following sentence to the manuscript: ‘’The preliminary search strategy will be applied without modification during the final search as it remained appropriate.’’

- Although it is mentioned that the complete strategy is in Appendix 2, the main text could indicate: the use of Boolean operators; the searched fields (ti, ab, kw); and the cross-database adaptation strategy.

The following sentences have been added to the manuscript for further clarification:

‘’The Boolean operator OR was used to combine synonymous terms and closely related concepts within the search strategy.’’

‘’MeSH terms were used where available to improve search precision.’’

‘’A cross-database adoption strategy will be used to ensure a comprehensive and systematic identification of relevant literature.’’

---

## [Decision Letter · Decision Letter 1]

1 Feb 2026

Consumers’ Psychological Constructs regarding Hybrid Meat Products: a Scoping Review Protocol

PONE-D-25-63842R1

Dear Dr. Pekdemir,

We’re pleased to inform you that your manuscript has been judged scientifically suitable for publication and will be formally accepted for publication once it meets all outstanding technical requirements.

Kind regards,

António Raposo

Academic Editor

PLOS One

Additional Editor Comments (optional):

Reviewers' comments:

Reviewer's Responses to Questions

**Comments to the Author**

1. Does the manuscript provide a valid rationale for the proposed study, with clearly identified and justified research questions?

Reviewer #1: Yes

2. Is the protocol technically sound and planned in a manner that will lead to a meaningful outcome and allow testing the stated hypotheses?

Reviewer #1: Yes

3. Is the methodology feasible and described in sufficient detail to allow the work to be replicable?

Reviewer #1: Yes

4. Have the authors described where all data underlying the findings will be made available when the study is complete?

Reviewer #1: Yes

5. Is the manuscript presented in an intelligible fashion and written in standard English?

Reviewer #1: Yes

You may also provide optional suggestions and comments to authors that they might find helpful in planning their study.

Reviewer #1: After reviewing the revised version of the manuscript, I consider that the previously made suggestions have been adequately addressed. At this time, I have no further contributions to add. Congratulations!

**Do you want your identity to be public for this peer review?** For information about this choice, including consent withdrawal, please see our Privacy Policy

Reviewer #1: **Yes:** Rodrigo Assis Neves Dantas Dantas

---

## [Editor Report · Acceptance letter]

PONE-D-25-63842R1

PLOS One

Dear Dr. Pekdemir,

I'm pleased to inform you that your manuscript has been deemed suitable for publication in PLOS One. Congratulations! Your manuscript is now being handed over to our production team.

Kind regards,

on behalf of

Dr. António Raposo

Academic Editor

PLOS One